# Comparative Analysis of Microbial and Mycotoxin Contamination in Korean Traditional Soybean Paste and Soy Sauce Production with and without Starter

**Jinkwi Kim [1,2,†], Jiyoun Jeong [1,3,†], Mi Jang [1], Jong-Chan Kim [1,*] and Heeyoung Lee [1,*]**

[1]  Food Standard Research Center, Korea Food Research Institute, Wanju 55365, Republic of Korea; wlsrnl1970@naver.com (J.K.); allenjeong@kfri.re.kr (J.J.); jangmi@kfri.re.kr (M.J.)
[2]  Department of Food Science and Technology, Chung-Ang University, Anseong 17546, Republic of Korea
[3]  Department of Industrial Systems Engineering, Jeonbuk National University, Jeonju-si 54896, Republic of Korea
*   Correspondence: jckim@kfri.re.kr (J.-C.K.); hylee06@kfri.re.kr (H.L.); Tel.: +82-63-9155 (J.-C.K.); +82-63-9454 (H.L.)
†   These authors contributed equally to this work.

**Abstract:** *Doenjang* and *ganjang* are traditional, Korean fermented foods. They are manufactured by fermenting jangs, either through the traditional natural fermentation, using straw, or the commercial inoculating starter cultures. However, both *Bacillus cereus* and aflatoxins have been detected in jangs, compromising their safety. Aflatoxins have been detected on numerous occasions. However, studies are yet to be conducted on whether these safety issues differ depending on the manufacturing method. In this study, we evaluated whether the manufacturing methods of *doenjang* and *ganjang* alter their safety. Samples of traditional and commercial *doenjang* and *ganjang* were analyzed for aflatoxin and B. cereus contamination. Microbiome taxonomic profiling was performed to assess microbial composition. The experimental methodology involved sample collection from various stages of production, including the use of starter cultures and natural fermentation processes. Aflatoxin levels were determined using regulatory limits, and B. cereus content was evaluated against specific thresholds. Aflatoxins were detected in both traditional and commercial *doenjang* and *ganjang*, with either the total aflatoxin (15 μg/kg) or aflatoxin B1 (10 μg/kg) exceeding the regulatory limits. However, ochratoxin A was not detected in any of the samples. *B. cereus* was detected in some samples, within the regulatory limit (4 log CFU/g), and was not influenced by the manufacturing method. Analysis at the production stage showed that aflatoxin increased alongside the fermentation time in traditional *doenjang*. However, in *ganjang*, no significant differences were associated with the fermentation period. When using starter cultures, the fermentation period did not affect the toxin level in both foods. Both methods showed lower aflatoxin content in the initial *doenjang* and *ganjang* samples than in *meju*. *B. cereus* was not detected in either method, as its content decreased over the fermentation period. Microbiome taxonomic profiling confirmed that even when using starter cultures, *B. cereus* was not a dominant species and was considerably affected by the environment. *Staphylococcus aureus* and *Pseudomonas*, pathogenic in nature, were detected in products manufactured using the traditional method; hence, the potential risk of this method was higher than that of the commercial method. The experimental methodology employed in this study contributes to understanding the microbial composition and toxin contamination levels in *doenjang* and *ganjang*, contributing to the overall knowledge of their safety and quality control.

**Keywords:** *doenjang*; *ganjang*; mycotoxin; *Bacillus cereus*; manufacturing methods; traditional fermentation



## 1. Introduction

*Doenjang* and *ganjang* are fermented foods made from soybeans and are traditionally consumed in Korea. These products are often used in seasoning for their specific flavor and taste. Soybeans have a high protein content (38% protein) and represent a major source of protein in the grain-oriented diets common to Asia [1]. Boiled soybeans are used to

make *meju*. This is then soaked in brine, the liquid of which is separated and processed into *ganjang*. The resulting solid can be ripened further to produce *doenjang* [2–4].

There are two main methods used to manufacture these fermented products. The traditional method of *meju* manufacture involves weaving rice straw, whose naturally occurring bacteria contribute to the fermentation process. As a result, *meju* manufactured using natural fermentation intrinsically contains a variety of microorganisms. Owing to the metabolism of these microorganisms during fermentation and maturation, the resulting food product exhibits the unique taste and aroma of soybean paste. However, the combination of the microorganisms naturally found in the straw and the long manufacturing period renders challenges in controlling the growth of microorganisms during fermentation, giving rise to numerous potential safety issues [5,6].

Therefore, the identification of the strains involved in the fermentation of jang is an important topic of study. Owing to the advances in science and technology, the manufacture of jang by fermenting *meju* via seed treatment without a straw is gaining popularity. This method, known as the commercial method, involves inoculating a starter culture or using koji, which can be produced using the starter culture. Since the fermentation period of the commercial method is shorter than that of the traditional method, standardization is easier, and the production of products of a certain quality, in addition to the advantage of controlling the safety of the products, is possible [7,8].

The long manufacturing period of *doenjang* and *ganjang* increases the chances of contamination by microorganisms and mycotoxins, which can be harmful to humans when ingested. To assess food safety, the microbiological quality of food must be analyzed. The presence of pathogenic organisms such as *S. aureus*, *Escherichia coli*, and *B. cereus* in food is a cause of concern as it can lead to several foodborne diseases, as well as food poisoning [9]. At present, quantitative standards have only been introduced for the presence of *B. cereus* in soybeans in Korea. Owing to the lack of established standards for other harmful microorganisms, a need to investigate the degree of contamination of other microorganisms, in addition to *B. cereus*, has gained significance.

Mycotoxins are secondary metabolites produced by mold that cause diseases and abnormal physiological effects in humans and livestock. Among these, aflatoxin is produced by fungi of the *Aspergillus* genus. Aflatoxins (AFs) are comprised of over 20 types of toxins, among which aflatoxin B1 (AFB1) has been reported as the most toxic, inducing liver cirrhosis and liver cancer. Aflatoxin is classified as a Group 1 (carcinogenic to humans) carcinogen, as designated by the World Health Organization's (WHO) International Agency for Research on Cancer (IARC) [10–12]. Ochratoxin is another secondary metabolite produced by fungi such as *Aspergillus ochraceus* and *Penicillium verrucosum*, and is a known carcinogen. The most prevalent and toxic ochratoxin is ochratoxin A (OTA). The IARC is responsible for identifying and classifying substances that are potentially carcinogenic to humans (Group 2B) (IARC, 2020). Studies have found that aflatoxin and ochratoxin have optimal growth conditions at 30 °C and 15–30 °C, respectively [13,14]. Notably, the optimum temperature for mold growth is similar to the temperature of *meju* during fermentation (15–30 °C). Contaminated *meju* carries a high risk of mold growth during the manufacturing process, and therefore the regulation of the process is required to mitigate this risk. In Korea, the regulatory limits for total AF, AFB1, and OTA are 15 μg/kg, 10 μg/kg, and 20 μg/kg, respectively.

Previous studies have reported the presence of harmful microorganisms, such as *B. cereus*, and mycotoxins, particularly aflatoxins, in fermented food products, exceeding regulatory limits. Researchers have aimed to enhance the productivity, safety, and quality of *doenjang* and *ganjang* production in response. However, limited research exists on the effect of using starters, or of their absence, on the safety of soybean paste and soy sauce. Therefore, this study aims to conduct a comparative analysis of microbial and mycotoxin contamination based on the use of starters in the production of traditional Korean soybean paste and soy sauce.

## 2. Materials and Methods

### 2.1. Sample Preparation

Supplementary Figure S1 presents the traditional preparation method for soy sauce and soybean paste, along with the commercial method that utilizes starters [2]. Monitoring samples were collected through visits to markets, online retailers, and companies. The samples were then divided into groups for processing using either the commercial method, using starter cultures, or the traditional method of natural fermentation, which involved the use of starter cultures, or the traditional method of natural fermentation, which involved the use of straw. A total of 24 samples of domestic *doenjang* and *ganjang* were obtained for this study. To examine the influence of starter culture on the manufacturing process of *doenjang* and *ganjang*, both methods with and without starter culture were employed for direct production. The production stage sample was obtained from a company located in Sunchang-gun, Jeollabuk-do. Among the samples used in the previous experiment, two companies employing the commercial method and two companies employing the traditional method were selected. All *meju* was manufactured in November, and samples were collected for each production cycle. Experiments were conducted based on these samples.

### 2.2. Measuring Aflatoxins and Ochratoxin A in Doenjang and Ganjang

#### 2.2.1. Standards and Reagents

High-performance liquid chromatography (HPLC) grade water, acetonitrile (CAN), and methanol were obtained from Avantor (Radnor, Pennsylvania, USA), ensuring their high purity and quality. Phosphoric acid and trifluoroacetic acid (TFA) were purchased from Sigma-Aldrich (St. Louis, MO, USA) and used as mobile phase additives for chromatographic separation. Aflatoxin standard mixtures (B1, B2, G1, and G2) were purchased from Romer Labs (Tulln, Austria) to serve as calibration standards. Ochratoxin A standard was obtained from Sigma-Aldrich (St. Louis, MO, USA) and dissolved in acetonitrile to prepare the corresponding standard solution. To maintain their stability, all standard solutions were carefully stored at $-20$ °C until further analysis.

#### 2.2.2. Sample Preparation

The sample (4 g of doenjang, 6 mL of water, 10 g of ganjang) was accurately weighed and transferred to a 50 mL centrifuge tube to initiate the extraction process. To facilitate mycotoxin extraction, 10 mL of 1% formic acid in acetonitrile, QuEChERS powder, and two ceramic homogenizers were added to the sample. The mixture was vigorously shaken for 1 minute, ensuring proper extraction. Subsequently, centrifugation was performed at 4000 rpm for 5 minutes to separate the supernatant.

Following centrifugation, 5 mL of the acetonitrile (ACN) supernatant was carefully transferred to a dSPE tube (dispersive solid-phase extraction) and subjected to another round of shaking for 1 minute. After an additional centrifugation step at 4000 rpm for 5 minutes, 2 mL of the purified supernatant was collected and transferred to a 2 mL EP tube.

To concentrate the sample, the ACN extract was evaporated using a speed Vac apparatus under air-dry conditions at 40 °C. Subsequently, 500 µL of rehydration solution (20% ACN:TFA [8:2]) was added to the concentrated sample. The mixture was allowed to react in the dark for 20 minutes, followed by filtration using a 0.45 µm filter [15].

Finally, the analysis was conducted using high-performance liquid chromatography (HPLC) with fluorescence detection, utilizing the conditions described in Supplementary Table S1.

### 2.3. Measuring Microbial Contamination Levels in Doenjang and Ganjang

For microbiological analysis, 30 g of each sample was collected, and to initiate the dilution process, 90 g of sterile water was added, resulting in a peptone water dilution. To detect aerobic bacteria, 1 mL of the pre-treated sample was inoculated on a dry film medium for aerobic bacteria (3M, Saint Paul, MI, USA) and incubated at 37 °C for 48 h. The subsequent step involved counting the number of red colonies generated. Similarly,

to detect *E. coli*, 1 mL of the pre-treated sample was inoculated on a dry film medium for *E. coli* (3M, Saint Paul, MI, USA) and incubated at 37 °C for 48 h. The colonies that formed bubbles around the newly created blue colonies were counted.

For coliform detection, 1 mL of the pre-treated sample was inoculated on a dry film medium for coliforms (3M, Saint Paul, MI, USA) and incubated at 37 °C for 24 h. The resulting red colonies and colonies that formed bubbles were counted. For *B. cereus* identification, 0.1 mL of the diluted sample was spread on the surface of a mannitol-egg-yolk-polymyxin agar (MYP) plate (Merck, Darmstadt, Germany) and incubated at 30 °C for 24 h, after which the resulting colonies were counted. Lastly, for *S. aureus* analysis, 0.1 mL of each sample was spread on the surface of a Baird Parker agar (BPA) plate (Merck, Darmstadt, Germany) and incubated at 37 °C for 24 h. The colonies exhibiting typical *S. aureus* morphology were counted.

### 2.4. Microbial Taxonomic Profiling

Total DNA was extracted using the FastDNA® SPIN Kit for Soil (MP Biomedicals, Irvine, CA, USA), in accordance with the manufacturer's instructions. Polymerase chain reaction (PCR) amplification was performed using the extracted DNA and fusion primers targeting the V3 to V4 regions of the 16S rRNA gene. To amplify the bacteria, fusion primers 341F and 805R were used. The fusion primers were constructed in the following order: P5 (P7) graft binding, i5 (i7) index, Nextera consensus, sequencing adaptor, and target region sequence. The amplifications were performed under the following conditions: initial denaturation at 95 °C for 3 min, followed by 25 cycles of denaturation at 95 °C for 30 s, primer annealing at 55 °C for 30 s, and extension at 72 °C for 30 s, with a final elongation at 72 °C for 5 min. The PCR product was confirmed using 1% agarose gel electrophoresis and visualized using a Gel Doc system (BioRad, Hercules, CA, USA). The amplified products were purified using CleanPCR (CleanNA). Equal concentrations of purified products were pooled together, and short fragments (non-target products) were removed using CleanPCR (CleanNA). The quality and product size were assessed on a Bioanalyzer 2100 (Agilent, Palo Alto, CA, USA) using a DNA 7500 chip. Mixed amplicons were pooled, and the sequencing was carried out at Chunlab, Inc. (Seoul, Korea), using the Illumina MiSeq Sequencing system (Illumina, San Diego, CA, USA) according to the manufacturer's instructions. The EzBioCloud 16S rRNA database was used for a taxonomic assignment using the usearch_global command of VSEARCH, followed by a more precise pairwise alignment. Chimeric reads were filtered on reads with <97% similarity with reference-based chimeric detection using UCHIME algorithm6 and the non-chimeric 16S rRNA database from EzBioCloud. After chimeric filtering, reads that were not identified to the species level (with <97% similarity) in the EzBioCloud database were compiled, and cluster_fast command2 was used to perform de novo clustering to generate additional OTUs. The sequenced data were analyzed using the EzBioCloud database.

## 3. Results and Discussion

### 3.1. Monitoring the Risk Factors of Commercial Products according to the Manufacturing Methods

3.1.1. Detection of Aflatoxins and Ochratoxin A in Doenjang and Ganjang

As shown in Table 1, in traditional *doenjang*, 1.64–87.67 μg/kg of total aflatoxins was detected in 14 samples (60.9%). In addition, 1.43–8.45 μg/kg of aflatoxin B1 was detected in 12 samples (52.2%). The mean concentration was 3.84 and 8.22 for AFB1 and total AFs, respectively, in the samples. One sample (4.3%) exceeded the regulatory limits for total AFs (15 μg/kg). In the commercial *doenjang*, aflatoxin was detected in three samples (16.7%) out of eighteen samples, and AFB1 was present in the range of 3.14–16.68 μg/kg, with an average of 10.20 μg/kg. Among these, two samples (11.1%) exceeded the regulatory limits for AFB1 (10 μg/kg). Aflatoxins were detected in 12 samples (54.5%) of traditional *ganjang* and were contaminated to 0.20–35.75 μg/kg. The AFB1 content was 3.39–5.07 μg/kg in one sample (9.1%). The average detection concentration was 4.33 μg/kg for AFB1 and 7.80 μg/kg for total AFs, and three samples (13.6%) exceeded the regulatory limits for

total AFs (15 μg/kg). In commercial *ganjang*, aflatoxin was detected in 13 (76.5%) out of 17 samples and was detected at a level of 1.57–14.24 μg/kg only in AFB1. It was detected at an average concentration of 6.57 μg/kg, and exceeded the regulatory limit (10 μg/kg) in four samples (23.5%). Ochratoxin A was not detected in any sample. The manufacturing method did not significantly affect contamination in the commercial product.

**Table 1.** Occurrence of aflatoxins and ochratoxin A in *doenjang* and *ganjang*.

|  | Sample | Number of Samples | Number of Positive Samples | Range | | | | Mean |
|---|---|---|---|---|---|---|---|---|
|  |  |  |  | AFB1 (ug/kg) | AFG1 (ug/kg) | Total Afs (ug/kg) | OTA (ug/kg) | Total Afs (ug/kg) |
| *Doenjang* | Traditional | 24 | 14 | 1.43–8.45 | 2.10–87.67 | 1.64–87.67 | ND | 8.22 |
|  | Commercial | 24 | 12 | 3.14–16.68 | 0.25–35.75 | 0.25–35.75 | ND | 10.20 |
| *Ganjang* | Traditional | 18 | 3 | 3.39–5.07 | ND | 3.14–16.68 | ND | 7.80 |
|  | Commercial | 17 | 13 | 1.57–14.24 | ND | 1.57–14.24 | ND | 6.57 |

ND: Not detected.

3.1.2. Measuring Microbial Contamination Levels in *Doenjang* and *Ganjang*

As shown in Table 2, the results of the microbial analysis of 20 samples of *doenjang* did not indicate any incidence of *E. coli* or *E. coli* groups. As for harmful microorganisms, *S. aureus* colony was not found, but *B. cereus* colony was detected. Therefore, we performed a test for *B. cereus*. *Ganjang* was not analyzed because it involves sterilization or heating in the production process, with little risk of microbial contamination [2,16]. *B. cereus* was detected in the range of 0–2.49 log CFU/g in five samples (20.8%) of traditional *doenjang*. In commercial *doenjang*, it was detected in the range of 0.81–3.23 log CFU/g in three samples (16.7%). In traditional *ganjang*, it was detected in the range of 0–0.62 log CFU/g in two samples (8.3%). In commercial *ganjang*, *B. cereus* was not detected in any of the 17 samples. The use of starter cultures did not create a significant difference in either *doenjang* or *ganjang*, and all samples were within the regulatory limits (4 log CFU/g) and evaluated to be safe.

**Table 2.** *Bacillus cereus* contamination levels in *doenjang* and *ganjang*.

|  | Sample | No. of Samples | No. of Positive Samples | Range (LogCFU/g) | Positive Mean (LogCFU/g) |
|---|---|---|---|---|---|
| *Doenjang* | Traditional | 24 | 5 | 0–2.49 | 0.58 |
|  | Commercial | 24 | 3 | 0.81–3.23 | 1.79 |
| *Ganjang* | Traditional | 18 | 2 | 0–0.62 | 0.18 |
|  | Commercial | 17 | 0 | ND | ND |

ND: Not detected.

*3.2. Analysis of Risk Factors according to the Production Stages*

3.2.1. Detection of Mycotoxins according to the Production Stage

The microbial community can change during fermentation, subsequently affecting the final product. Therefore, we evaluated the degree of mycotoxin contamination according to the production stage of *doenjang* and *ganjang*. As a result, ochratoxin A was not detected in any stage. Aflatoxins were not detected in soybeans from traditionally manufactured products. However, in *meju*, it was detected at the levels of 2.19 ± 0.08 and 1.34 ± 0.56 μg/kg during the fermentation period, which slightly decreased as the fermentation progressed. In *doenjang*, aflatoxin levels were 0.74 ± 0.35, 2.03 ± 0.74, and 5.32 ± 0.16 μg/kg during the fermentation period, indicating that mycotoxins increased as the aging period increased. In *ganjang*, no significant difference was observed during the fermentation period (0.24 ± 0.10, 0.26 ± 0.09, 0.22 μg/kg). Aflatoxins were not detected in soybeans from commercial products. Significant differences were not observed in *meju* (1.07 ± 0.26 and 1.11 ± 0.10 μg/kg) during the fermentation period. In the case of *doenjang*,

0.56 ± 0.07, 1.1 ± 0.18, and 1.11 ± 0.04 µg/kg were found, indicating an increase over the fermentation period. In *ganjang*, the toxin level gradually decreased to 0.97 ± 0.07, 0.6, and 0.3 ± 0.01 µg/kg (Figure 1). In both methods, the aflatoxin content was lower in the initial *doenjang* and *ganjang* samples than in *meju*. The reduction in toxin levels can be attributed to the inability of the mold to grow after soaking in brine during the ripening process [17,18].

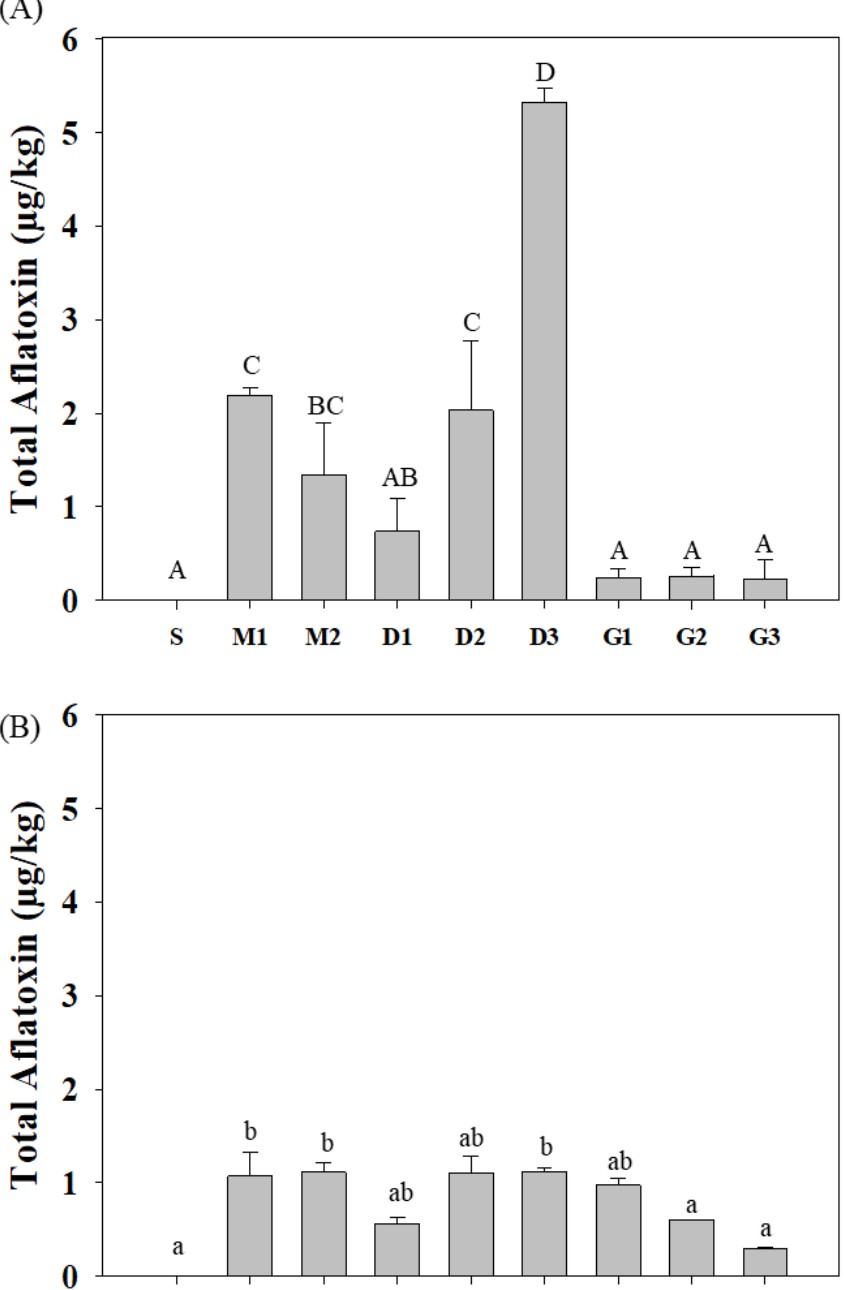

**Figure 1.** Total aflatoxin contamination levels at the production stage in (**A**) traditional method and (**B**) commercial method (S: soybean, M1: *meju*, M2: fermented *meju*, D1: *doenjang*, D2: *doenjang* fermented for 2 months, D3: *doenjang* fermented for 4 months, G1: *ganjang*, G2: *ganjang* fermented for 2 months, and G3: *ganjang* fermented for 4 months). Significance ($p < 0.05$) are presented as different letters.

### 3.2.2. Occurrence of B. cereus according to the Production Stage

The level of *B. cereus* in soybean was $0.63 \pm 0.01$ in the traditional method, while the levels were $4.32 \pm 0.30$ and $1.10 \pm 0.57$ µg/kg in *meju*, indicating a decrease with aging. In the initial *doenjang* sample, $2.61 \pm 0.47$ µg/kg was found, but *B. cereus* was not detected after ripening. In the initial *ganjang* sample, $0.36 \pm 0.12$ µg/kg was found, but it was not detected after aging. The amount of *B. cereus* was $3.25 \pm 0.15$ µg/kg in soybeans fermented using the commercial method and only 2.1 µg/kg in the initial *meju* samples. In the initial *doenjang* samples, $2.37 \pm 0.79$ µg/kg was found and was not detected thereafter. By contrast, *B. cereus* was not detected in *ganjang* at all (Figure 2).

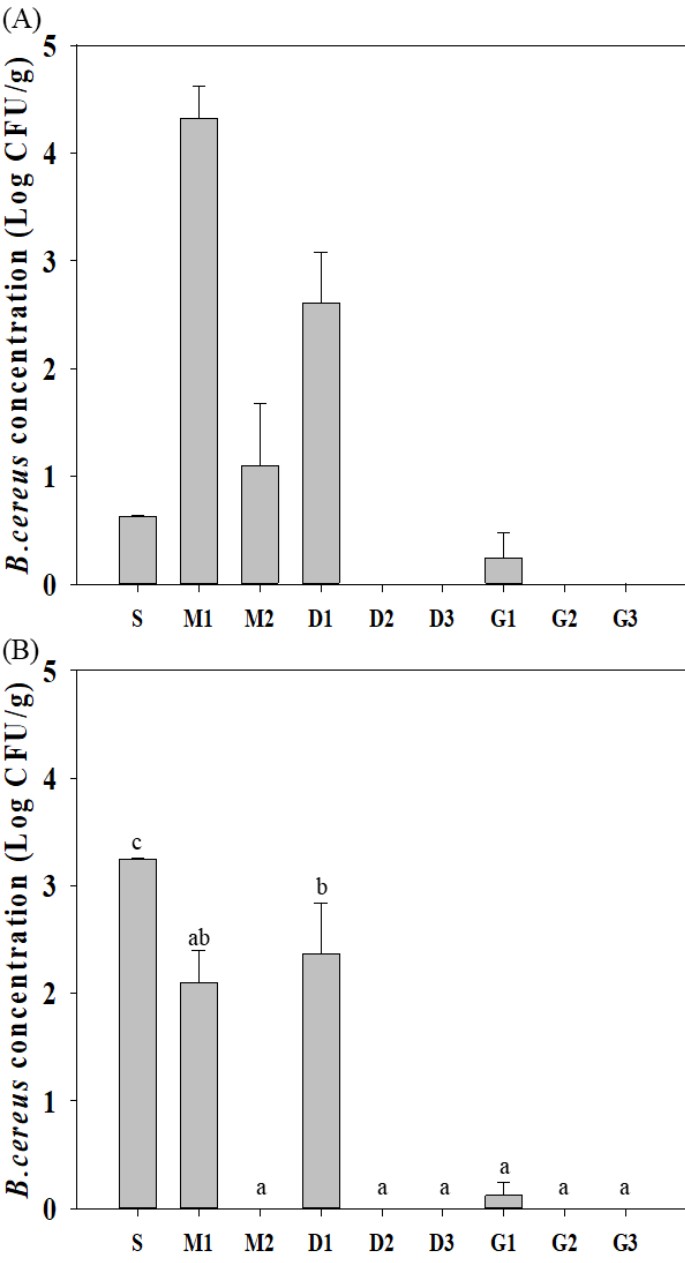

**Figure 2.** *B. cereus* contamination levels at the production stage in (**A**) traditional method and (**B**) commercial method (S: soybean, M1: *meju*, M2: fermented *meju*, D1: *doenjang*, D2: *doenjang* fermented 2 months, D3: *doenjang* fermented 4 months, G1: *ganjang*, G2: *ganjang* fermented 2 months, and G3: *ganjang* fermented 4 months). Significance ($p < 0.05$) are presented as different letters.

### 3.2.3. Microbial Taxonomic Profiling

To analyze the microbial communities, companies A and B used the traditional method, while companies C and D used the commercial method (Figure 3). At the species level, *Leuconostoc lactis* (77.7%) and *Leuconostoc citreum* (4.2%) were found in *meju*, while *Pseudomonas putida* (27.74%) and *Weissella confusa* (20.46%) were found in brine for company A. *Leuconostoc mesenteroides* (35.41%) and *Chromohalobacter marismortui* (29.42%) were dominant in *doenjang*. After ripening in *doenjang*, *C. marismortui* (35.34%) and *Chromohalobacter beijerinckii* (25.12%) were found to be dominant, in this order. In *ganjang*, *C. marismortui* (39.45% and 47.63%) and *C. beijerinckii* (26.63% and 29.10%) were abundant. In company B, *Bacillus licheniformis* (41.1%) and *Bacillus thermoamylovorans* (21.84%) were dominant in *meju*, and *C. beijerinckii* (52.84%) and *Tetragenococcus halophilus* (15.87%) were dominant in the brine. *T. halophilus* (48.8%) and *Bacillus licheniformis* (14.7%) were the dominant species in *doenjang* before ripening. After ripening, *S. aureus* (27.9%) and *T. halophilus* (14.7%) were found to be the most dominant. *C. beijerinckii* (52.82% and 38.92%) and *T. halophilus* (14.49% and 16.98%) were abundant in both *ganjang* and fermented *ganjang*. *Halomonas halmophila* (27.74%) was also abundant in fermented *ganjang*. In company C, we found *Enterococcus faecium* (33.8% and 55.0% and 61.7%) to be dominant in *meju* and *doenjang*, followed by *W. confusa* (26.7% and 13.9%), while *Pediococcus acidilactici* (22.4%) was found in fermented *doenjang*. In brine, *Staphylococcus succinus* (52.18%) and *Kushneria konosiri* (33.82%) were abundant. *Staphylococcus saprophyticus* (57.90%) was dominant in *ganjang*, followed by *W. confusa* (15.72%). *P. acidilactici* (26.29%) and *W. confusa* (21.41%) were abundant in fermented *ganjang*. In company D, we found *E. faecium* to be abundantly present at concentrations of 51.85%, 7.74%, 55.88%, 76.24%, and 24.99% in *meju*, brine, *doenjang*, fermented *doenjang*, and *ganjang*, respectively. *W. confusa* (43.1% and 54.91%) was abundant in *meju* and *ganjang*, while *P. acidilactici* (39.4% and 16.0%) was abundant in *doenjang*. In brine, *Bacillus subtilis* (61.87%) was the dominant species, and *Lentibacillus kimchi* (90.59%) was the most dominant in fermented *ganjang*.

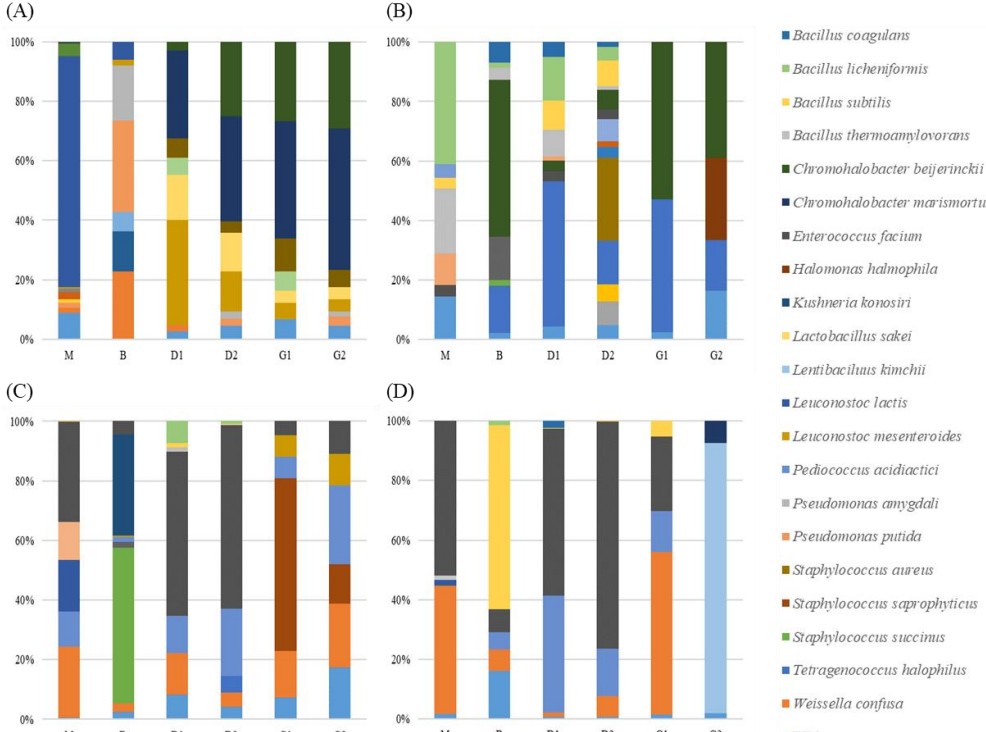

**Figure 3.** Microbial taxonomic classification at the species level. (**A**,**B**) Traditional methods; (**C**,**D**) commercial methods. (M: *meju*, B: brine, D1: *doenjang*, D2: fermented *doenjang*, G1: *ganjang*, and G2: fermented *ganjang*).

Companies C and D used *A. oryzae* and *B. subtilis* in starter cultures. However, none of the strains in the starter culture appeared as the dominant species in either of the samples produced by these two companies. Although microbial diversity was found to be reduced when the starter culture was used, its use can be considered to be advantageous owing to the absence of the pathogenic bacteria *S. aureus* and *Pseudomonas*. The overall microbial distribution appeared to be greatly influenced by the environment. Although the microbial distribution of the starter cultures used by the companies was similar, additional studies are needed to confirm these results owing to the small number of comparative samples included in the present study.

## 4. Conclusions

In conclusion, the two manufacturing methods did not create a significant difference in the toxin levels of *doenjang* and *ganjang*. However, as the fermentation period increased in the traditional process of *doenjang* preparation, aflatoxin levels increased. In addition, by confirming the presence of *Pseudomonas* and the pathogenic bacteria *S. aureus* in the product made using the traditional method, it is ascertained that the potential risk of this method is high; however, additional studies are required to authenticate the findings.

**Supplementary Materials:** The following supporting information can be downloaded at: https://www.mdpi.com/article/10.3390/fermentation9070621/s1. Supplementary Figure S1. Methods for the manufacture of Doenjang and Ganjang. Supplementary Table S1. Conditions for HPLC-FLD.

**Author Contributions:** Conceptualization, J.K., J.-C.K., M.J. and H.L.; data curation, J.K., J.J., M.J. and H.L.; formal analysis, J.K. and H.L.; methodology, J.-C.K. and H.L.; writing—original draft preparation, J.K. and H.L.; writing—review and editing, J.J. and J.-C.K. All authors have read and agreed to the published version of the manuscript.

**Funding:** This work was supported by the Study on Standardization for Korean Traditional Foods (G0221400-01) of the National Agricultural Products Quality Management Service (NAQS).

**Institutional Review Board Statement:** Not applicable.

**Informed Consent Statement:** Not applicable.

**Data Availability Statement:** Not applicable.

**Acknowledgments:** The authors thank the support of the Study on Standardization for Korean Traditional Foods of the National Agricultural Products Quality Management Service (NAQS).

**Conflicts of Interest:** There are no conflicts of interest to declare.

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
