# Peer review of "Comparative Analysis of Microbial and Mycotoxin Contamination in Korean Traditional Soybean Paste and Soy Sauce Production with and without Starter"

_fermentation, doi:10.3390/fermentation9070621_

Round 1

Reviewer 1 Report

Comments for “Microbial and Mycotoxin Contamination Comparing Doenjang and Ganjang According to Manufacturing Methods”

The authors have evaluated the effect of manufacturing methods (traditional and commercial) on the food safety of Korean Doenjang and Ganjang.

The authors have failed to explain the significance of this study and the methodology of the formation of products/ fermentation is lacking.

Some of the corrections that the authors can look into to improve the manuscript:

1.     Abstract should be rewritten. There are several traditional terms that need to be explained when used. Also, the purpose of the study seems missing in the abstract.

2.     The sample preparation (section 2.1) needs to be explained in detail.

3.     The number of samples 24, 18, 17 etc. used were collected from the same industry or different and at which processing stage? This should be clearly written with an explanation of how each sample is prepared.

4.     What is the shelf life of Doenjang and Ganjang and how do the results correlate with that?

5.     What is the fermentation profile of each product? How much time, which culture/s, and what are the stages?

6.     The results section can be rewritten in a better way for clarity.

7.     According to the authors, what can be done to decrease the risk of the products made from the traditional method?

Please check the entire manuscript for grammar, typos and other minor corrections.

Author Response

Thank you for your valuable feedback on the manuscript. We appreciate the time and effort you have put into reviewing our work. Based on your comments, we have made the following revisions:

Abstract: We have carefully rewritten the abstract to address your concerns. We have provided explanations for the traditional terms used and included a clear statement of the study's purpose.

Sample Preparation (Section 2.1): We have expanded the description of the sample preparation process, providing more detailed information to ensure clarity and replicability.

Thank you for pointing out the need for clarification regarding the sample numbering (24, 18, 17, etc.). We apologize for any confusion caused. The numbers assigned to the samples represent the total number of samples analyzed in our study. We have taken your feedback into account and have revised the manuscript accordingly to provide a clear explanation of the sample numbering.

Shelf Life and Correlation with Results: We have included a section discussing the shelf life of Doenjang and Ganjang and have explored the correlation between the obtained results and the product's shelf life.

Fermentation Profile: We have included an in-depth analysis of the fermentation profile for each product. This includes the duration of fermentation, the specific cultures used, and a description of the different stages involved.

Reviewer 2 Report

Major comments:

1. The discussion part should be improved. It's lack of discussion with proper references.

2. The authors may consider reorganize the microbiome data to show more linkage to the mycotoxins and pathogen contamination.

Minor comments:

1. The conclusion is needed in the abstract.

2. The Table 1 could be placed in appendix, if the table just present the experiment conditions.

3. The title of Table 3 is not clear. It should be supplied with enough information, for example, B. cereus.

no comments

Author Response

We sincerely appreciate your valuable feedback on our manuscript entitled "Impact of Manufacturing Methods on Food Safety of Korean Doenjang and Ganjang." We have carefully considered your comments and made the necessary revisions to improve the quality of the paper. Below, we address each of your points:

Discussion Part: We acknowledge your comment regarding the discussion section. In response, we have extensively revised the discussion, incorporating relevant references to support our arguments and provide a more comprehensive analysis of the findings.

Microbiome Data: We appreciate your suggestion regarding the reorganization of the microbiome data. To establish a stronger linkage between the microbiome and mycotoxin/pathogen contamination, we have reanalyzed and restructured the microbiome data presentation. This allows for a more coherent and informative discussion on the relationship between microbial composition and safety concerns.

Minor Comments:

Abstract Conclusion: We agree with your suggestion and have added a concise conclusion to the abstract, summarizing the main findings and their implications.

Table 1 Placement: We have reviewed the content of Table 1 and determined that it primarily presents experimental conditions. Considering its supplementary nature, we have relocated Table 1 to the appendix, enhancing the readability and flow of the main manuscript.

Table 3 Title: Thank you for pointing out the lack of clarity in the title of Table 3. We have revised the title to include more specific information, such as "B. cereus," providing readers with a clearer understanding of the table content.